# TiO$_2$-SnS$_2$ Nanoheterostructures for High-Performance Humidity Sensor

**Wencheng Yu [1], Duo Chen [1,2,*], Jianfei Li [1,*] and Zhenzhen Zhang [1,2,*]**

[1] International School for Optoelectronic Engineering, Qilu University of Technology (Shandong Academy of Science), Jinan 250353, China

[2] Laser Institute, Shandong Academy of Sciences, Jinan 250100, China

* Correspondence: drdchen@qlu.edu.cn (D.C.); jfl@qlu.edu.cn (J.L.); zzz@sdlaser.cn (Z.Z.)

**Abstract:** The larger surface-to-volume ratio of the hierarchical nanostructure means it has attracted considerable interest as a prototype gas sensor. Both TiO$_2$ and SnS$_2$ can be used as sensitive materials for humidity sensing with excellent performance. However, TiO$_2$-SnS$_2$ nanocomposites are rarely used in humidity detection. Therefore, in this work, a new humidity sensor was prepared by a simple one-step synthesis process based on nano-heterostructures, and the humidity sensing performance of the device was systematically characterized by much faster response/recovery behavior, better linearity and greater sensitivity compared to pure TiO$_2$ or SnS$_2$ nanofibers. The enhanced sensitivity of the nanoheterostructure should be attributed to its special hierarchical structure and TiO$_2$-SnS$_2$ heterojunction, which ultimately leads to a significant change in resistance upon water molecule exposure. In consideration of its non-complicated, cost-effective fabrication process and environmental friendliness, the TiO$_2$-SnS$_2$ nanoheterostructure is a hopeful candidate for humidity sensor applications.

**Keywords:** humid sensor; TiO$_2$-SnS$_2$ nanoheterostructure; composite





## 1. Introduction

In recent years, semiconductor composites have become one of the hottest topics all over the world. When two different semiconductors are combined together, some novel properties may appear. TiO$_2$-MoS$_2$ [1] and TiO$_2$-Sn$_3$O$_4$ [2] show an enhanced photocatalytic activity. TeO$_2$-SnO$_2$ [3] and CuO-ZnO [4] have improved gas-sensing properties. Similarly, TiO$_2$-SnO$_2$ [5] and TiO$_2$-ZnO [6] display excellent ultraviolet responsivity. Metal oxides are stable in structure; easy to synthesize; have good application prospects in photoelectric measurement, gas detection, etc.; and are a class of cost-effective materials. Among metal oxide semiconductors, titanium dioxide (TiO$_2$) is one of the most widely used wide-band-gap oxide semiconductors. It has excellent physical and chemical properties and low prices and has been widely used in practical life. Since Fujisima and Honda published their work on the catalytic water splitting of TiO$_2$ under ultraviolet light irradiation in the 1970s [7], the application of TiO$_2$ has been rapidly extended to the fields of optoelectronics [8], photocatalysis [9], photo/electrochromic [10] and gas detection [11]. Moreover, TiO$_2$ has good chemical stability and controllable morphology and is a good matrix material [12]. As an n-type semiconductor material, SnS$_2$ belongs to layered metal sulfide with a hexagonal CdI$_2$ crystal structure, and it is a novel two-dimensional material [13]. SnS$_2$ has a wide energy band gap (about 2.18 eV [13]) and strong anisotropic optical properties. Therefore, SnS$_2$ is often used in gas-sensitive materials [14], photoelectric equipment [15], optical materials [16] and other fields. TiO$_2$ and SnS$_2$ have been studied as active materials in humidity sensors. A humidity sensor based on sol–gel-prepared TiO$_2$ film has been reported by Giampiero Montesperelli et al., which shows high humidity sensitivity at the minimum relative humidity (RH) values (4–10% RH) at 40 °C [17]. Lakshmi Deepika Bharatula et al. demonstrated a SnS$_2$ nanoflake micro-nano sensor device that can work within the scope of

11–97% RH at room temperature [18]. $TiO_2$-$SnS_2$ nanocomposites are generally used as photocatalytic materials. For example, Marin Kovacic et al. used $TiO_2$-$SnS_2$ nanocomposites as solar-active photocatalytic materials for water treatment [19]. Then, in the field of humidity sensors, carbon-based nanomaterials are commonly used as sensing materials. Due to the large specific surface area of carbon nanotubes, they have good adsorption characteristics for water molecules and are good moisture-sensitive materials for humidity sensors, and they also meet the development trend of the integration and miniaturization of humidity sensors. Hai M. Duong et al. used a high-temperature CVD furnace to produce continuous macroscopic fibers and films made from CNT superfibers and used them in the field of humidity sensing [20]. In addition, Hai Minh Duong et al. found that since the properties of post-treated CNT fibers are comparable to many commercial high-strength fibers, such as carbon fiber T300, Dyneema and Twaron, they can be utilized as reinforcements for advanced composites. Nanotube-based composites made from unstructured CNT powder have been extensively applied as structural materials for a wide range of applications, such as automotive and aerospace applications [21]. Therefore, carbon-based nanomaterials also have a wide range of promising applications.

In the investigations to date, many $TiO_2$ and $SnS_2$-related composites have been used in humidity sensors. Dongzhi Zhang et al. fabricated a humidity sensor based on a $WS_2$/$SnO_2$ nanocomposite with improved sensitivity and rapid response compared to pure $WS_2$ and pure $SnO_2$, and it also performs quite well in detecting human respiration [22]. Dongzhi Zhang et al. prepared $SnS_2$/$Zn_2SnO_4$ hybrid spherical films as sensitive materials for humidity sensors utilizing a layer-by-layer self-assembly technique. They found that the $SnS_2$/$Zn_2SnO_4$ hybrid thin-film sensors made significant progress in humidity sensors compared to single-$SnS_2$ and single-$Zn_2SnO_4$ nanomaterials, achieving accurate measurements of human breath, sweat, urine and water droplets [23]. Yun Wang et al. successfully fabricated tubular $TiO_2$-$SnO_2$ fibers (FIT-TSF) using a general crystal-phase-induced formation strategy. The prepared FIT-TSF exhibited excellent sensing performance with third-order impedance variation, an ultra-fast response time of 5 s, a recovery time of 8 s and good reproducibility [24]. Irene Cappelli et al. analyzed the performance of different humidity sensors based on $TiO_2$ nanoparticles and correlated them with different chemical/physical phenomena, and they found that when relative humidity is greater than 70%, the presence of condensate changes the electrical properties of the sensor, resulting in a smaller equivalent resistance value and a larger equivalent capacitance value of the sensor. The sensor has the advantages of a relatively fast response, a large measurement range and good stability [25]. However, there are few studies using $TiO_2$-$SnS_2$ nanocomposites for humidity detection.

In this paper, we synthesized high-quality $TiO_2$ nanoribbons by the hydrothermal method and acid treatment, and we dispersed $SnS_2$ nanoparticles on $TiO_2$ nanoribbons to form $TiO_2$-$SnS_2$ nanoheterostructures. The morphology and structure of bare $TiO_2$ nanoribbons and $SnS_2$-$TiO_2$ nanoheterostructures were characterized by transmission electron microscopy, scanning electron microscopy, Raman spectrum and X-ray diffraction. The optical properties of the bare $TiO_2$ and $TiO_2$-$SnS_2$ nanoheterostructures were characterized by reflection spectroscopy. The humidity sensors were prepared using bare $TiO_2$ nanoribbons, $SnS_2$ nanoparticles and $TiO_2$-$SnS_2$ nanoheterostructures as active materials, and their humidity detection performance at room temperature was investigated. Finally, through comparative experiments, we found that the resistance changes of the detector based on $TiO_2$-$SnS_2$ are linear with the relative humidity, while the resistance change of the two detectors based on pure $TiO_2$ and pure $SnS_2$ is not linear in the process of humidity change. In addition, the resistance of the two detectors based on pure $TiO_2$ and pure $SnS_2$ can reach $10^{10}$ ohms under low relative humidity, which is difficult to measure accurately. The resistance of $TiO_2$-$SnS_2$ is only in the order of kiloohms, which can be easily detected with an ordinary multimeter. This is coupled with the fact that the synthesis process of $TiO_2$-$SnS_2$ nanoheterostructures in this work is simpler and less costly than that of general metal

oxide composites. Therefore, the humidity sensor based on a $TiO_2$-$SnS_2$ nanostructure is more suitable for daily applications.

## 2. Materials and Methods

Analytically pure Titania P25 ($TiO_2$: ca. 80% anatase (CAS. 13463-67-7) and 20% rutile (CAS. 1317-80-2)), sodium hydroxide (NaOH (CAS. 1310-73-2)), hydrochloric acid (HCl (CAS. 7647-01-0)), sulfuric acid ($H_2SO_4$ (CAS. 7664-93-9)), tin(IV) chloride ($SnCl_4 \cdot 5H_2O$ (CAS. 10026-06-9)) and thioacetamide (CAS. 62-55-5) were used without further purification. A homogeneous solution was made by mixing 0.8 g of P25 $TiO_2$ with 80 mL of aqueous 10 M NaOH. The mixture was then shifted to a 100 mL Teflon (CAS. 9002-84-0) stainless steel autoclave and heated at 180 °C for 72 h. $Na_2Ti_3O_7$ nanoparticles were gained after thoroughly washing the obtained powder with deionized water. $H_2Ti_3O_7$ nanospheres were produced by immersing 0.47 g of $Na_2Ti_3O_7$ nanospheres into 58.8 mL of 0.1 M hydrochloric acid for 24 h. Finally, $H_2Ti_3O_7$ nanospheres (0.285 g) were etched in 14.25 mL of 0.02 M $H_2SO_4$ aqueous solution at 100 °C for 2 h to obtain rough nanospheres. The products were separated from the solution by centrifugation, washed thoroughly with deionized water in turn, and then annealed at 600 °C for 2 h to obtain $TiO_2$ nanospheres.

$TiO_2$-$SnS_2$ nanoheterostructures were prepared by a simple hydrothermal co-precipitation method. In a typical process, 2.5 mmol $SnCl_4 \cdot 5H_2O$ and 25 mmol thioacetamide were dissolved in 18 mL of deionized water to make a transparent solution, to which a certain amount of pre-synthesized $TiO_2$ nanobelts (molar ratio of Sn/Ti = 1/1) was added. Then, the solution was injected into a 20 mL Teflon-lined stainless-steel autoclave and maintained at 160 °C for 12 h. The obtained $TiO_2$-$SnS_2$ nanoheterostructures were washed with deionized water and dried at 70 °C. Bare $SnS_2$ was also prepared using a similar method.

The synthesized powders were dispersed in deionized water. Then, several drops of the obtained suspension were directly dropped onto a precleaned alumina (CAS. 1344-28-1) substrate, followed by thermal annealing at 100 °C for 30 min. Finally, interlaced gold electrodes (width: 100 μm; pitch: 200 μm) were deposited on the sample surface by thermal evaporation for humidity detection measurements.

The crystal structure of the samples was examined by X-ray diffraction (XRD, XD-3, PG Instruments Ltd., Beijing, China) and Raman (Bruker, Ltd., Billerica, MA, USA). The surface morphology of the samples was characterized by using field emission scanning electron microscopy (SEM; Hitachi S-4800, Hitachi, Ltd., Chiyoda, Tokyo, Japan). Transmission electron microscopy (TEM) and scanning transmission electron microscopy (STEM) were collected on a JEOL JEM 2100F electron microscope (JEM 2100F, JEOL Ltd., Tokyo, Japan) with an accelerating voltage of 200 kV. We used a saturated salt solution humidity generator to measure the humidity sensitivity of the sample. LiCl- (CAS. 7447-41-8), $MgCl_2 \cdot 6H_2O$- (CAS. 7791-18-6), NaBr- (CAS. 7647-15-6), NaCl- (CAS. 20510-56-9), KCl- (CAS. 7447-40-7) and $KNO_3$- (CAS. 14797-55-8) saturated salt solutions, whose corresponding humidity at room temperature was 11.30%, 32.78%, 57.57%, 75.29%, 84.34% and 93.58%, respectively, were chosen as humidity generators. Taking the relative humidity of the lithium-chloride-saturated salt solution as the background humidity, the response of $TiO_2$, $SnS_2$ and $TiO_2$-$SnS_2$ was measured under different moderate conditions.

## 3. Results and Discussion

As illustrated in Figure 1, the crystal structure of the synthetic $TiO_2$ nanobelts and $TiO_2$-$SnS_2$ nanoheterostructures was investigated by XRD. Figure 1a depicts the XRD pattern of $TiO_2$ nanobelts, with all diffraction peaks matching those of anatase $TiO_2$ (JCPDS card, no. 21-1272). For the $TiO_2$-$SnS_2$ (Figure 1b), besides the diffraction peaks from anatase $TiO_2$, all other peaks can be indexed to hexagonal-structured $SnS_2$ (JCPDS card, no. 23-0677). Figure 2 shows the Raman spectrums of $TiO_2$, $SnS_2$ and $TiO_2$-$SnS_2$. For the $TiO_2$, as shown in Figure 2a, three distinct peaks, centered at 237, 252 and 294 $cm^{-1}$, can be observed. Figure 2b shows the Raman spectral lines of the $SnS_2$ that has one distinct peak, centered at 310 $cm^{-1}$. The red curve in Figure 2 shows that the $TiO_2$-$SnS_2$ has three peaks centered at

235, 257 and 285 cm$^{-1}$, which roughly correspond to the three distinct peaks of TiO$_2$, and has one peak centered at 309 cm$^{-1}$, corresponding to the peak of SnS$_2$. No unambiguous signal from others is observed. These results confirm the successful deposition of SnS$_2$ on TiO$_2$ nanobelts, which is further proved by SEM and TEM in the following part.

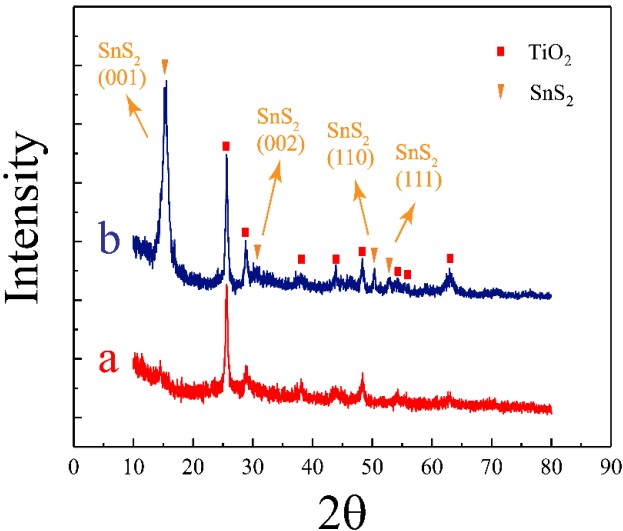

**Figure 1.** XRD patterns of (**a**) the TiO$_2$ belts and (**b**) the TiO$_2$-SnS$_2$ nanostructure.

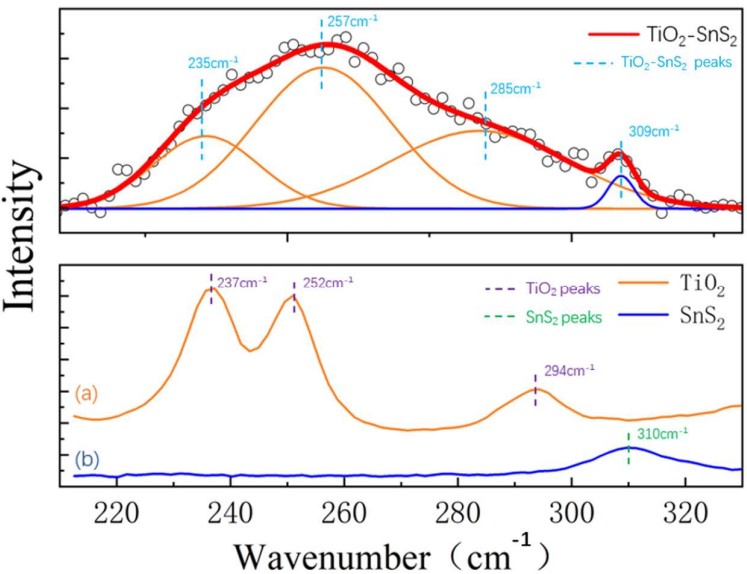

**Figure 2.** Raman spectrums of (**a**) the TiO$_2$, (**b**) the SnS$_2$ and the TiO$_2$-SnS$_2$.

Figure 3 shows the SEM images of the TiO$_2$ nanobelts and the TiO$_2$-SnS$_2$ nanoheterostructures. The surface of the TiO$_2$ nanobelts is relatively smooth, and the nanobelts are straight, as shown in Figure 3a. A higher-magnification SEM image (Figure 3b) shows that there are many randomly distributed dark depressions on the nanobelts as a result of the acid etching of the nanobelts. The morphology of the TiO$_2$-SnS$_2$ is shown in Figure 3c. The surface of the TiO$_2$ nanobelts is rough due to the deposition of SnS$_2$ nanoparticles. For better observation, a single-TiO$_2$ nanobelt decorated with SnS$_2$ is checked under high magnification, as shown in Figure 3d. It can be observed that the SnS$_2$ nanoparticles are scattered on the TiO$_2$ nanoribbons in a random shape.

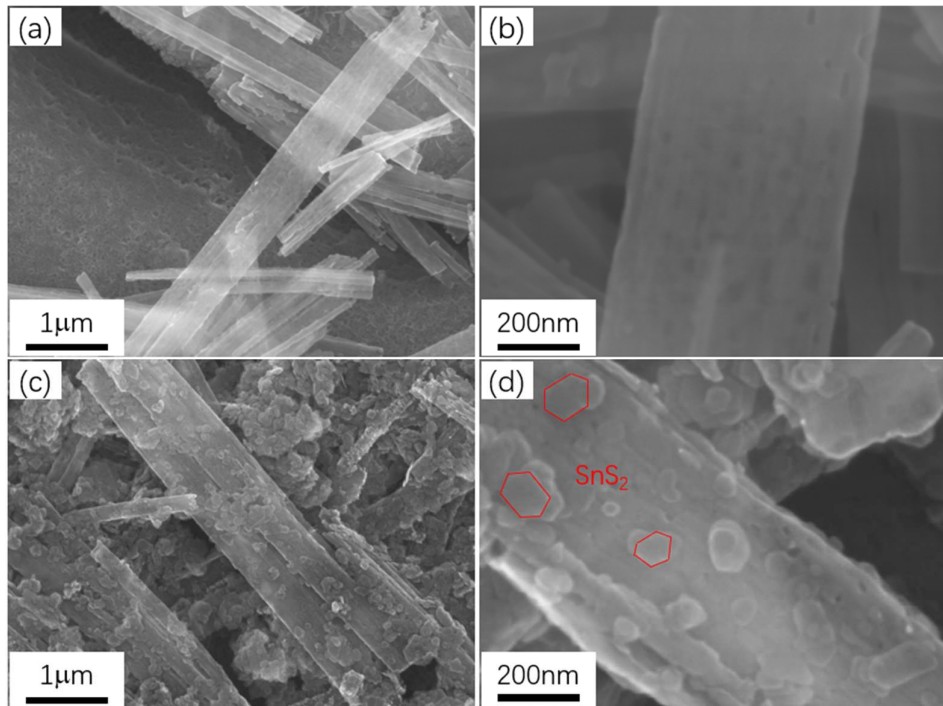

**Figure 3.** SEM images of (**a**,**b**) the TiO$_2$ nanobelts and (**c**,**d**) the TiO$_2$-SnS$_2$.

The morphology of the TiO$_2$ nanobelts is characterized by TEM, as shown in Figure 4a. The nanobelts are uniform and straight, with widths of 60–180 nm. A higher-magnification TEM image (Figure 4b) shows that there are many bright and dark regions randomly distributed on the nanobelts, which indicates that the nanobelt surface becomes rough due to acid etching. However, the single-crystal structure of TiO$_2$ is reserved even after acid etching, as demonstrated by HRTEM and the electron diffraction pattern in Figure 4c. The morphology of the TiO$_2$-SnS$_2$ nanostructure is shown in Figure 4d. SnS$_2$ nanoparticles are deposited on the surface of the TiO$_2$ nanobelt evenly. In order to facilitate better observation, a SnS$_2$-decorated single-TiO$_2$ nanobelt is checked at high magnification, as shown in Figure 4e. It can be seen that the size of SnS$_2$ nanoparticles is small and relatively homogeneous. As exhibited by HRTEM analysis of the TiO$_2$-SnS$_2$ nanoheterostructure in Figure 4f, SnS$_2$ nanoparticles with a size of approximately 10 nm are distributed on the TiO$_2$ nanobelt uniformly and densely.

Figure 5a shows the optical reflection spectra of bare TiO$_2$ and TiO$_2$-SnS$_2$ nanoheterostructures. A sharp decrease in the reflectivity can be observed at approximately 560 nm for TiO$_2$-SnS$_2$ and 370 nm for TiO$_2$, which can be attributed to the interband absorption of TiO$_2$-SnS$_2$ [26,27] and anatase TiO$_2$ [28]. The TiO$_2$-SnS$_2$ nanostrcuture exhibits lower reflection than the bare TiO$_2$ in visible light, which comes from the increased light absorption by SnS$_2$ nanoparticles [29]. Figure 5b shows the plots of $(F(R)E)^2$ as a function of photon energy E for TiO$_2$-SnS$_2$ and TiO$_2$ samples (F(R) is the Kubelka–Munk function; $F(R) = (1-R)^2/2R$, where R is the reflectance). The band gap can be determined by the linear extrapolation of $(F(R)E)^2$ to 0. The deduced band gap is about 3.36 eV for TiO$_2$ and 2.31 eV for TiO$_2$-SnS$_2$. Because of its strong absorption in visible light, the TiO$_2$-SnS$_2$ nanostructure may be used as the active layer in a visible-light photodetector.

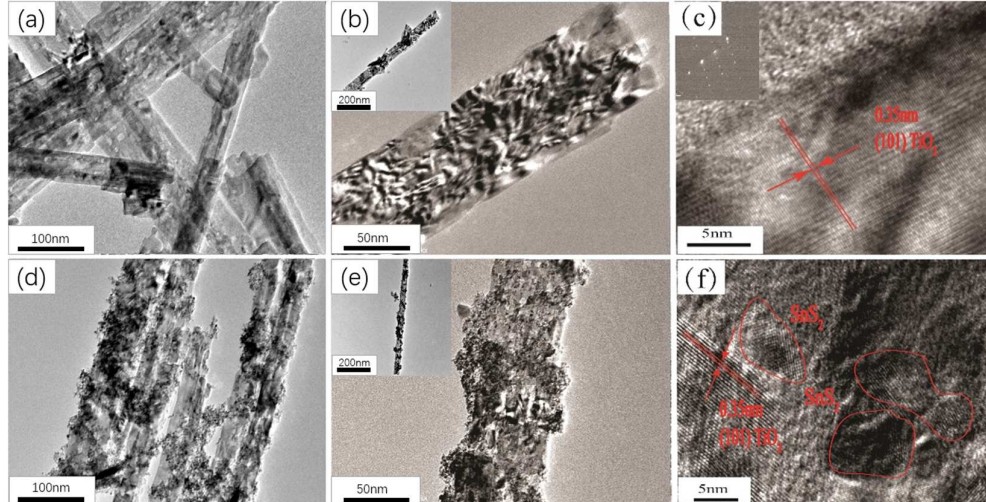

**Figure 4.** TEM images (**a**,**b**) and HRTEM image (**c**) of the TiO$_2$ nanobelts. TEM images (**d**,**e**) and HRTEM image (**f**) of TiO$_2$-SnS$_2$. Insets: (**b**) TEM image of single-TiO$_2$ nanobelt, (**c**) the electron diffraction pattern image of TiO$_2$ and (**e**) TEM image of single-SnS$_2$-TiO$_2$ nanoheterostructure.

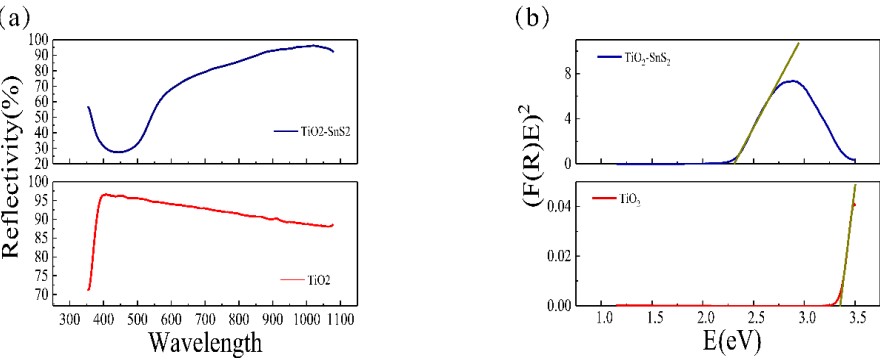

**Figure 5.** (**a**) The reflection spectra of TiO$_2$-SnS$_2$ and TiO$_2$ nanobelts. (**b**) Plots of $(F(R)E)^2$ versus photon energy E.

A saturated salt solution humidity generator, also known as a fixed-point humidity generator, has lots of advantages, such as simple equipment, cost-effectiveness, stable humidity value, easy recovery after damage, good reproducibility and so on. In this article, the humidity sensitivity of samples is measured using the saturated salt solution humidity generator. LiCl-, MgCl$_2$·6H$_2$O-, NaBr-, NaCl-, KCl- and KNO$_3$-saturated salt solutions, whose corresponding humidity at room temperature is 11.30%, 32.78%, 57.57%, 75.29%, 84.34% and 93.58%, respectively, are chosen as humidity generators. The relative humidity of LiCl-saturated salt solution is taken as the background humidity, and the response curves of TiO$_2$, SnS$_2$ and TiO$_2$-SnS$_2$ at different humidity levels are shown in Figure 6. As shown in Figure 6a, the resistance of the SnS$_2$-TiO$_2$-based sensor is much smaller than that of the bare TiO$_2$ or SnS$_2$-based sensor, and the change in resistance from RH 11% to RH 93% reaches three orders of magnitude. However, in low relative humidity, the TiO$_2$ nanobelt device is in a high-resistance state (Figure 6b), and the resistance value is too high for an ordinary instrument. The response at different RH of the SnS$_2$ device is shown in Figure 6c. The resistance of this device is basically unchanged at lower RH, and the resistance change reaches three orders of magnitude from low RH to high RH. Figure 6d–f show the resistance of TiO$_2$-SnS$_2$, TiO$_2$ and SnS$_2$ sensors at different RH from 32.78% to 93.58%. The resistance change of the TiO$_2$-SnS$_2$ device is close to a linear relation, while the performance of the other two devices is unsatisfactory. Furthermore, the resistance of TiO$_2$ or SnS$_2$ devices reaches 10$^{10}$ ohm, which is hard to be detected by ordinary instruments. In comparison,

the resistance of $TiO_2$-$SnS_2$ is only in the order of kiloohm, which can be easily detected by a multimeter. In other words, the device based on $TiO_2$-$SnS_2$ nanostructures is more practical for daily-life applications.

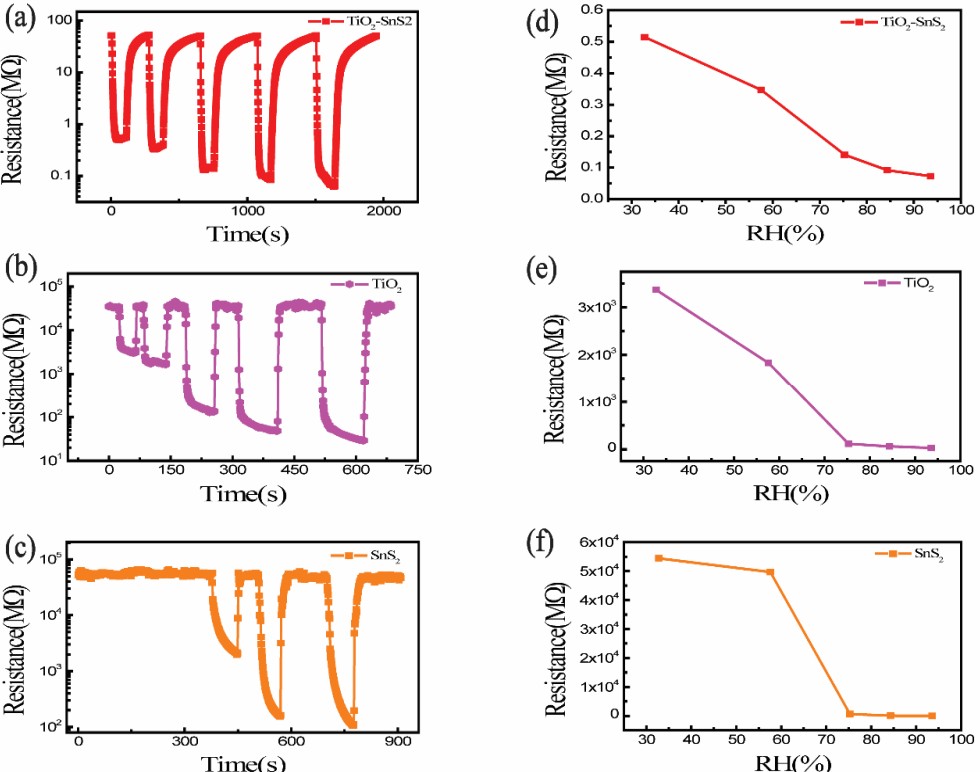

**Figure 6.** The humid responsivity (**a**) and (**d**) at different RH of $TiO_2$-$SnS_2$, the humid responsivity (**b**) and (**e**) at different RH of $TiO_2$, and the humid responsivity (**c**) and (**f**) at different RH of $SnS_2$.

Pure $TiO_2$ and $SnS_2$ have different responses to humidity, and the composite of the two materials $TiO_2$-$SnS_2$ shows different response mechanisms. First of all, we describe the sensing mechanisms of monolithic components (i.e., pure $TiO_2$ and $SnS_2$). The moisture sensing of $TiO_2$ is mainly caused by oxygen vacancies [30], and compounds containing alkali ions usually show significant hydrophilic properties due to their surface alkalinity [31]. When $H_2O$ combines with variations, the conductivity is promoted. For $SnS_2$, the sensing mechanism here is dominant by proton conduction [32]. If abundant water molecules were adsorbed at the $SnS_2$ surface, proton conduction would be formed. As Figure 7a shows, when $SnS_2$ (band gap $E_g$ = 2.18 eV [13]) loads on $TiO_2$ ($E_g$ = 3.2 eV [33]), a potential barrier develops at the $TiO_2$-$SnS_2$ heterojunction. The equivalent resistance of the whole system is the series resistance of the $R_t$, $R_s$ and $R_h$. $R_t$ means the resistance of $TiO_2$, $R_s$ means the resistance of $SnS_2$, and $R_h$ means the resistance of the heterojunction. When the sensor is exposed to a low relative humidity environment, water molecules are adsorbed on the $TiO_2$ surface. It reduces the $R_t$. Therefore, equivalent resistance reduces. As the relative humidity increases, water molecules are further adsorbed due to the electrostatic effect of the OH- groups, and a physical adsorption water layer is formed [34]. Protons transfer from water molecules to $SnS_2$, and the potential barrier height reduces further, as shown in Figure 7b. The layer facilitates the transfer of $H_2O$ or $H_3O^+$ [35,36]. The quick transfer of ions in the aqueous layer significantly decreases the impedance, which gives rise to the high sensitivity of the sensor.

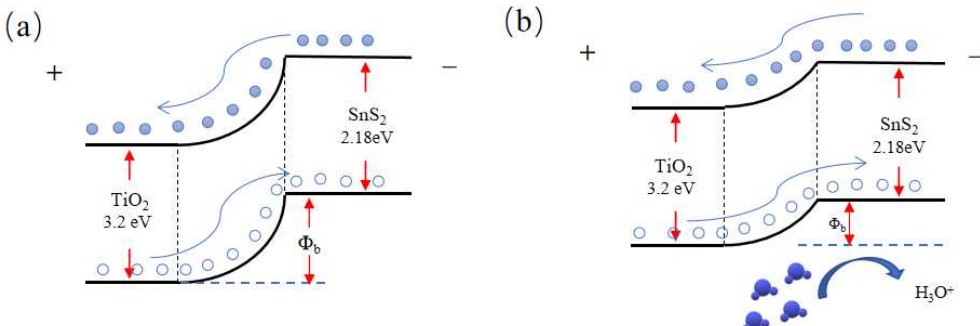

**Figure 7.** The energy band variations of the $TiO_2$-$SnS_2$ heterojunction after adsorbing water molecules at (**a**) without water and (**b**) with water.

Table 1 shows a comparison of the performance of some humidity sensors using various materials as sensitive materials. The main comparisons are made in terms of the fabrication method, measuring range, and response time. From the aspect of the fabrication method, this work uses a one-step hydrothermal method to prepare the samples, and the production process is relatively simple. From the aspect of measuring range, other parts of the material are similar to the samples prepared in this work and all have a large measurement range. From the aspect of response time, there exist other sensitive materials with more rapid response times, but the response time of the samples in this work is also faster. Dongzhi Zhang et al. used a simple one-step hydrothermal route for the preparation of microelectrode polyimide substrates to synthesize $SnO_2$ nanoparticles and $SnO_2$/RGO hybrids using hydrothermal methods and used them as sensitive materials to fabricate humidity sensors [37]. The sensor has the advantages of high sensitivity and fast measurement speed, but processing the material into a sensor is relatively tedious. Ravindra Kumar Jha and Prasanta Kumar Guha synthesized $WS_2$ nanosheets in a binary mixture of acetone and acetone by ultrasound and used them as a sensing material for a humidity sensor [38]. The sensor has a good linear relationship with the response of humidity, and the repeatability and stability are also good, but the disadvantage of this sensor is that the measurement range is relatively small. Ming-Zhi Yang et al. fabricated an integrated humidity microsensor using a commercial 0.18 μm complementary metal oxide semiconductor (CMOS) process [39]. The advantage of this sensing is the synthesis of a miniature humidity sensor that is very convenient, but one of the biggest limitations of this sensor is that the measurement range is too small. Yinghua Tan et al. prepared hollow $MoS_2$ micro@nano-sphere composites by a one-step hydrothermal method and used this material as a sensitive material to prepare humidity sensors [40]. They found that the sensor has high sensitivity and good stability through implementation, but the sensor has a relatively small measurement range compared to our humidity sensor. Hui Yang et al. prepared capacitive humidity sensors by sequentially coating aqueous suspensions of zinc oxide (ZnO) nanopowders and polyvinylpyrrolidone-reduced graphene oxide (PVP-RGO) nanocomposites dropwise on cross-finger electrodes [41]. The sensor showed a significant improvement in sensitivity and linearity compared with PVP RGO/ZnO, PVP-RGO and ZnO for the ZnO/PVP-RGO sensor. However, the preparation process of this sensor material is more tedious than the preparation process of the sensitive material used in this paper. Hengchang Bi et al. fabricated a miniature capacitive humidity sensor using a graphene oxide thin film as a humidity sensing material [42]. Compared with the conventional capacitive humidity sensor, this sensor has a high sensitivity at 15–95% relative humidity, which is more than 10 times more sensitive than the best of the conventional sensors. However, the fabrication process of this sensor is relatively tedious. Comparing all aspects, we can find that $TiO_2$-$SnS_2$ nanoheterostructures are indeed an excellent candidate to be used as sensitive materials for humidity sensors.

**Table 1.** Comparison of the main features of previously reported humidity sensors.

| Active Materials | Fabrication Method | Measuring Range | Response | Reference |
|---|---|---|---|---|
| Graphene/$SnO_2$ | Hydrothermal | 11–97%RH | 560.85 | [37] |
| $WS_2$ | Liquid exfoliation | 40–80%RH | 37.5 | [38] |
| ZnO | Sol–gel method | 40–90%RH | - | [39] |
| $MoS_2$ | Hydrothermal | 17.2–89.5%RH | 67.34 | [40] |
| ZnO/PVP/RGO | Drop-casting | 15–95%RH | - | [41] |
| Graphene oxide | Solution dripping | 15–95%RH | 378 | [42] |
| $TiO_2$-$SnS_2$ | Hydrothermal | 11–93%RH | 60 | This work |

From the comparison of experimental results, the resistance of the $TiO_2$-$SnS_2$-based sensor is much smaller than that of the bare $TiO_2$ or $SnS_2$-based sensor, and the change in resistance from RH 11% to RH 93% reaches three orders of magnitude, from 0.05 MΩ to 100 MΩ. However, at low relative humidity, the $TiO_2$ nanoribbon devices are in a high resistance state, which is too high for ordinary instruments. The resistance of the $SnS_2$ nanoparticle devices is essentially unchanged at lower relative humidity, and the change in resistance from low to high relative humidity reaches three orders of magnitude. Resistance of the $TiO_2$-$SnS_2$ device varies linearly with humidity, while the other two devices do not perform as well. In addition, the resistance of $TiO_2$ or $SnS_2$ devices reaches $10^{10}$ ohms, which is difficult to detect with ordinary instruments. In contrast, the resistance of $TiO_2$-$SnS_2$ is only kiloohms, which can be easily detected with a multimeter. In addition, the roughness of the $TiO_2$ surface also affects the performance of the humidity sensor, which can be expressed as the larger the specific surface area of $TiO_2$, the better the effect of the humidity sensor, and conversely, the smaller the specific surface area of $TiO_2$, the worse the effect of the humidity sensor [43]. In terms of the humidity sensing performance of sensitive materials, the humidity sensors based on $TiO_2$-$SnS_2$ nanostructures are more suitable for everyday applications.

## 4. Conclusions

In summary, high-quality rough $TiO_2$ nanobelts were synthesized, and $SnS_2$ nanoparticles were loaded on the $TiO_2$ nanobelts by a hydrothermal method. The humidity detectors were fabricated using the powders of $TiO_2$-$SnS_2$, $TiO_2$ and $SnS_2$. Our research results show that the measurement range of $TiO_2$-$SnS_2$ nanoheterostructures in humidity detection is 11–93%, the response time is 60 s, and the linearity between resistance and humidity is good. In addition, the preparation of $TiO_2$-$SnS_2$ nanoheterostructures only uses a one-step hydrothermal method, the preparation process is very simple and convenient, and the cost of preparation is relatively low. These advantages will further increase the possibility of $TiO_2$-$SnS_2$ nanoheterostructures being excellent candidates for humidity detectors.

However, there are still some limitations to the $TiO_2$-$SnS_2$ we have synthesized. The structure of the $TiO_2$-$SnS_2$ nanoheterostructure we synthesized now is not very regular, and the performance of the humidity sensor needs to be further improved. In the next research, we will enhance the structure and number of heterojunctions, further modulate the structural properties of the interface to improve the humidity response of the sensor, and gradually explore the specificity of the sensor for some other gases to increase the practicality of the sensor.

**Author Contributions:** Conceptualization, D.C. and W.Y.; methodology, D.C.; software, W.Y.; validation, D.C.; formal analysis, W.Y.; investigation, J.L.; resources, D.C.; data curation, W.Y.; writing-original draft preparation, D.C.; writing-review and editing, W.Y.; visualization, J.L.; supervision, Z.Z.; project administration, Z.Z.; funding acquisition, D.C. All authors have read and agreed to the published version of the manuscript.

**Funding:** This work was supported by the Shandong Provincial Natural Science Foundation (ZR2021QF133), Basic research projects of science, education and industry integration pilot projects of Qilu University of Technology (2022PX037).

**Data Availability Statement:** The data that support the findings of this study are available within this article.

**Acknowledgments:** The authors acknowledge the experimental support of the International School for Optoelectronic Engineering at Qilu University of Technology (Shandong Academy of Science).

**Conflicts of Interest:** The authors declare no conflict of interest.

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
