# Peer review of "TiO2-SnS2 Nanoheterostructures for High-Performance Humidity Sensor"

_crystals, doi:10.3390/cryst13030482_

Round 1

Reviewer 1 Report

In this paper Authors presented a humidity sensor based on nanoheterostructures. Sensor is fabricated by a facile one-step synthetic process and the humidity sensing performance of this device is characterized systematically. Below I presented some remarks that came to my mind during reading:

1.       In my opinion the Introduction must be improved. In a research paper, it is expected that introduction section briefly explains the starting background and, even more important, the originality (novelty) and relevancy of the study is well established. Introduction should adequately represent the state of knowledge and clearly specify the purpose and motivation of taking up the topic. The area of research must be introduced with details for unfamiliar readers. The Authors should state what is special, unexpected, or different in their approach. The Authors should perform a critical survey of what has been done up to this point in the scientific literature and identify a precise gap in the current state of knowledge that needs to be filled, a gap that is being addressed by their research.

2.       Can the Authors provide the CAS numbers of the used materials?

3.       In Discussion the Authors should highlight what are the advantages and disadvantages when comparing solutions from the scientific literature and also highlighting current limitations of their study. There is no reference in the text to other literary studies and a broader discussion of the obtained results, comparing them with other results (other researchers, etc.). From my point of view, it has no justification not conducting a comparison with similar works already published. Such comparison significantly raises the meaning of the presented paper. The Authors should highlight what are the advantages and disadvantages when comparing solutions from the scientific literature.

4.       Conclusions should be worded slightly different. Try to emphasize novelty. Put some quantifications, and comment on the limitations. This is a very common way to write conclusions for a learned academic journal. In the Conclusions also, it would be useful to add information on further research of the authors related to the continuation of this research topic.

Author Response

Dear Editor and reviewers,

   Thank you very much for the reviewers’ helpful comments and suggestions to our manuscript entitled “TiO2-SnS2 nanoheterostructures for high performance humidity sensor”. We have considered all the comments carefully and completed our modifications according to your suggestions.

The comments are answered point to point in the following:

Responses to Reviewer 1

  1. In my opinion the Introduction must be improved. In a research paper, it is expected that introduction section briefly explains the starting background and, even more important, the originality (novelty) and relevancy of the study is well established. Introduction should adequately represent the state of knowledge and clearly specify the purpose and motivation of taking up the topic. The area of research must be introduced with details for unfamiliar readers. The Authors should state what is special, unexpected, or different in their approach. The Authors should perform a critical survey of what has been done up to this point in the scientific literature and identify a precise gap in the current state of knowledge that needs to be filled, a gap that is being addressed by their research.

  Reply: Thanks for reviewer’s helpful suggestion. We have revised the introduction as requested, adding some of the critical investigation of the present work to date. The additional sections are as follows:

  “In the investigation studies to date, there are many TiO2 and SnS2 related compo-sites used in humidity sensors. Dongzhi Zhang, Yuhua Cao et. al. made a kind of humidity sensor based on WS2/SnO2 nanocomposite, which has a high sensitivity and fast response compared to pure WS2 and pure SnO2, and he also performs quite well in detecting human respiration. Dongzhi Zhang, Xiaoqi Zong et. al. prepared SnS2/Zn22SnO4 hybrid spherical films as sensitive materials for humidity sensors using a layer-by-layer self-assembly technique, and they found that the SnS2/Zn2SnO4 hybrid thin-film sensor made great progress in humidity sensing compared with single SnS2 and single Zn2SnO4 nanomaterials and achieved accurate measurements of human breath, sweat, urine, and water droplets. Yun Wang, Xiaojuan Zhao et. al. successfully fabricated tubular TiO2-SnO2 fibers (FIT-TSF) using a general crystal phase induced formation strategy. The prepared FIT-TSF exhibited excellent sensing performance with third-order impedance variation, ultra-fast response time of 5 s, recovery time of 8 s, and good reproducibility. Irene Cappelli, Ada Fort et. al. analyzed the performance of different humidity sensors based on TiO2 nanoparticles and correlated them with different chemical/physical phenomena, and they found that when relative humidity is greater than 70%, the presence of condensate changes the electrical properties of the sensor, resulting in a smaller equivalent resistance value and a larger equivalent capacitance value of the sensor.”

  1. Can the Authors provide the CAS numbers of the used materials?

  Reply: Thanks for reviewer’s helpful comment. We apologize for missing the CAS number of the material we used. We have added the CAS numbers of all the materials used in Section Materials and Methods.

  1. In Discussion the Authors should highlight what are the advantages and disadvantages when comparing solutions from the scientific literature and also highlighting current limitations of their study. There is no reference in the text to other literary studies and a broader discussion of the obtained results, comparing them with other results (other researchers, etc.). From my point of view, it has no justification not conducting a comparison with similar works already published. Such comparison significantly raises the meaning of the presented paper. The Authors should highlight what are the advantages and disadvantages when comparing solutions from the scientific literature.

  Reply: Thank reviewer for your suggestion. In part Discussion, we added additional studies on humidity sensors. We compared several different materials in various aspects and analyzed the strengths and weaknesses of this work by comparison. The additional parts are as follows:

  “Table 1 shows a comparison of the performance of some humidity sensors using various materials as sensitive materials. The main comparisons are made in terms of fabrication method, measuring range and response time. From aspect fabrication method, this work uses a one-step hydrothermal method to prepare the samples, and the production process is relatively simple. From aspect measuring range, other parts of the material are similar to the samples prepared in this work, and all have a large measurement range. From aspect response time, there exist other sensitive materials with more rapid response time, but the response time of the samples of this work is also faster. Dongzhi Zhang et.al. used a simple one-step hydrothermal route for the preparation of microelectrode polyimide substrates to synthesize SnO2 nanoparticles and SnO2/RGO hybrids using hydrothermal methods, and used them as sensitive materials to fabricate humidity sensors [37]. The sensor has the advantages of high sensitivity and fast measurement speed, but the process of processing the material into a sensor is relatively tedious. Ravindra Kumar Jha and Prasanta Kumar Guha synthesized WS2 nanosheets in a binary mixture of acetone and acetone by ultrasound, and used them as a sensing material for a humidity sensor [38]. The sensor has a good linear relationship with the response of humidity, and the repeatability and stability are also good, but the disadvantage of this sensor is that the measurement range is relatively small. Ming-Zhi Yang et.al. fabricated an integrated humidity microsensor using a commercial 0.18 μm complementary metal oxide semiconductor (CMOS) process [39]. The advantage of this sensing is the synthesis of a miniature humidity sensor that is very convenient, but one of the biggest limitations of this sensor is that the measurement range is too small. Yinghua Tan et.al. prepared hollow MoS2 micro@nano-spheres composites by a one-step hydrothermal method and used this material as a sensitive material to prepare humidity sensors [40]. They found that the sensor has high sensitivity and good stability through implementation, but the sensor has a relatively small measurement range compared to our humidity sensor. Hui Yang et.al. prepared capacitive humidity sensors by sequentially coating aqueous suspensions of zinc oxide (ZnO) nanopowders and polyvinylpyrrolidone reduced graphene oxide (PVP-RGO) nanocomposites dropwise on cross-finger electrodes [41]. The sensor showed a significant improvement in sensitivity and linearity compared with PVP RGO/ZnO, PVP-RGO and ZnO for the ZnO/PVP-RGO sensor. However, the preparation process of this sensor material is tedious than the preparation process of the sensitive material used in this paper. Hengchang Bi et.al. fabricated a miniature capacitive humidity sensor using graphene oxide thin film as a humidity sensing material [42]. Compared with the conventional capacitive humidity sensor, this sensor has a high sensitivity at 15%-95% relative humidity, which is more than 10 times more sensitive than the best of the conventional sensors. However, the fabrication process of this sensor is relatively tedious. Comparing all aspects, we can find that TiO2-SnS2 nanoheterostructures are indeed an excellent candidate to be used as sensitive materials for humidity sensors.”

Table 1. Comparison of the main features of previously reported humidity sensors.

Active materials

Fabrication method

Measuring range

Response

Reference

Graphene/SnO2

Hydrothermal

11−97%RH

560.85

[37]

WS2

Liquid exfoliation

40−80%RH

37.5

[38]

ZnO

Sol-gel method

40-90%RH

-

[39]

MoS2

Hydrothermal

17.2−89.5%RH

67.34

[40]

ZnO/PVP/RGO

Drop-casting

15-95%RH

-

[41]

Graphene oxide

Solution dripping

15−95%RH

378

[42]

TiO2-SnS2

Hydrothermal

11-93%RH

60

This work

Reference:

[37] Zhang, H. Chang, P. Li, R. Liu, Q. Xue, Fabrication and Characterization of an Ultrasensitive Humidity Sensor Based on Metal Oxide/Graphene Hybrid Nanocomposite, Sensors and Actuators B: Chemical 4005(15)30606-7.

[38] Ravindra Kumar Jha and Prasanta Kumar Guha, Liquid exfoliated pristine WS2 nanosheets for ultrasensitive and highly stable chemiresistive humidity sensors, Nanotechnology 27 (2016) 475503.

[39] Ming-Zhi Yang, Ching-Liang Dai, and Chyan-Chyi Wu, Sol-Gel Zinc Oxide Humidity Sensors Integrated with a Ring Oscillator Circuit On-a-Chip, Sensors 14 (2014) 20360-20371.

[40] Yinghua Tan, Ke Yu, Ting Yang, Qingfeng Zhang, Weitao Cong, Haihong Yin, Zhengli Zhang, Yiwei Chen and Ziqiang Zhu, The combinations of hollow MoS2 micro@nano-spheres: one-step synthesis, excellent photocatalytic and humidity sensing properties, J. Mater. Chem. C, 2 (2014) 5422.

[41] Hui Yang, Qiangqiang Ye, Ruixue Zeng, Junkai Zhang, Lei Yue, Ming Xu, Zhi-Jun Qiu and Dongping Wu, Stable and Fast-Response Capacitive Humidity Sensors Based on a ZnO Nanopowder/PVP-RGO Multilayer, Sensors 17 (2017) 2415.

[42] Bi, H., Yin, K., Xie, X. et al. Ultrahigh humidity sensitivity of graphene oxide. Sci Rep 3 (2013) 2714.

4. Conclusions should be worded slightly different. Try to emphasize novelty. Put some quantifications, and comment on the limitations. This is a very common way to write conclusions for a learned academic journal. In the Conclusions also, it would be useful to add information on further research of the authors related to the continuation of this research topic.

  Reply: Thanks to the reviewer for this valuable comment. We have revised the wording of our conclusions, emphasized our strengths and analyzed the limitations of this work, and added directions and priorities for our subsequent research. The changes made are as follows:

  “In summary, high quality rough TiO2 nanobelts were synthesized and SnS2 nano-particles were loaded on the TiO2 nanobelts by a hydrothermal method. The humidity detectors were fabricated using the powders of TiO2-SnS2, TiO2 and SnS2. Our research results show that the measurement range of TiO2-SnS2 nanoheterostructures in humidity detection is 11-93%, the response time is 60s, and the linearity between resistance and humidity is good. In addition, the preparation of TiO2-SnS2 nanoheterostructures uses only a one-step hydrothermal method, and the preparation process is very simple and convenient, and the cost of preparation is relatively low. These advantages will further increase the possibility of TiO2-SnS2 nanoheterostructures to be excellent candidates for humidity detectors.

  However, there are still some limitations for the TiO2-SnS2 we have synthesized. The structure of the TiO2-SnS2 nanoheterostructure we synthesized now is not very regular, and the performance of the humidity sensor needs to be further improved. In the next research, we will enhance the structure and number of heterojunctions, further modulate the structural properties of the interface to improve the humidity response of the sensor, and gradually explore the specificity of the sensor for some other gases to increase the practicality of the sensor.”

Reviewer 2 Report

The authors developed a novel humidity sensor based on nanohetero-structures by using hydrothermal co-precipitation method. The developed device exhibited good humidity sensing performance with faster response/recovery behavior, better linearity, and higher sensitivity as compared to the pure TiO2 or SnS2 nanofibers. The results suggest that the TiO2-SnS2 nanoheterostructure is a promising candidate for low-cost and high-performance humidity sensors. The work is interesting and can be published in Crystals if the following issues can be addressed:

1. The works of https://doi.org/10.1016/B978-0-12-812667-7.00001-X and https://doi.org/10.1016/B978-0-08-102722-6.00006-7 should be cited in the introduction for better review of the applications of carbon-based nanomaterials for humidity sensor applications. 

2.   The author mentioned that the SnS2 nanoparticles in Figure 3(d) are in a hexagonal shape scattered on the TiO2 nanobelt. However, the particles in the figure seem to have a random shape and very few hexagonal shapes is clearly observed. The author should clarify this in the revised manuscript.

3. Why does Figure 4 a) and d) have different scale bars? Scale bars for insets of Figure 4 b) and e) are required.

4. References are required for the claim of “The TiO2-SnS2 nanostrcuture exhibits lower reflection than the bare TiO2 at the visible light, which comes from the increased light absorption by SnS2 nanoparticles”

5. Does the rough surface of the TiO2 affect the sensing performance?

6. Some grammar errors in the manuscript need to be corrected.

Author Response

Dear Editor and reviewers,

     Thank you very much for the reviewers’ helpful comments and suggestions to our manuscript entitled “TiO2-SnS2 nanoheterostructures for high performance humidity sensor”. We have considered all the comments carefully and completed our modifications according to your suggestions.

     The comments are answered point to point in the following:

Responses to Reviewer 2

1. The works of https://doi.org/10.1016/B978-0-12-812667-7.00001-X and https://doi.org/10.1016/B978-0-08-102722-6.00006-7 should be cited in the introduction for better review of the applications of carbon-based nanomaterials for humidity sensor applications. 

  Reply: Thank reviewer for your suggestion. We have already cited these two works in the introduction. The specific quotes are as follows:

  “Then, in the field of humidity sensor, carbon-based nanomaterials are usually used as sensing materials. Due to the large specific surface area of carbon nanotubes, they have good adsorption characteristics for water molecules and are good moisture-sensitive materials for humidity sensors, and also meet the development trend of integration and miniaturization of humidity sensors. Hai M. Duong et. al. used a high-temperature CVD furnace to produce continuous macroscopic fibers and films made of CNT superfibers, and make it used in the field of humidity sensing [20]. And Hai Minh Duong et. al. found that as the performance of the posttreated CNT fibers was comparable to many commercial high-strength fibers, such as carbon fibers T300, Dyneema, and Twaron, they can be used as reinforcement for advanced composites. Nanotube-based composites fabricated from unstructured CNT powders have been widely used for a broad range of application, such as structural materials for automotive and aerospace applications [21]. Therefore, carbon-based nanomaterials also have a wide range of promising applications.”

Reference:

[20] Hai M. Duong, Thang Q. Tran, Reed Kopp, Sandar Myo Myint, Liu Peng, Direct Spinning of Horizontally Aligned Carbon Nanotube Fibers and Films From the Floating Catalyst Method, Nanotube Superfiber Materials (Second Edition), (2019) 3-29.

[21] Hai Minh Duong, Sandar Myo Myint, Thang Quyet Tran, Duyen Khac Le, Post-spinning treatments to carbon nanotube fibers, Carbon Nanotube Fibers and Yarns, (2020) 103-134.

2. The author mentioned that the SnS2 nanoparticles in Figure 3(d) are in a hexagonal shape scattered on the TiO2 nanobelt. However, the particles in the figure seem to have a random shape and very few hexagonal shapes are clearly observed. The author should clarify this in the revised manuscript.

  Reply: Thanks for reviewer’s helpful comment. We apologize for the lack of ambiguity in this statement. SnS2 nanoparticles are indeed in a random shape, and we have made modifications. The specific modifications are as follows:

   “It can be observed that the SnS2 nanoparticles are scattered on the TiO2 nanoribbons in a random shape.”

3. Why does Figure 4 a) and d) have different scale bars? Scale bars for insets of Figure 4 b) and e) are required.

 Reply: Thanks to the reviewer for this valuable comment. We are very sorry for our mistake. We have revised the figures. 

4. References are required for the claim of “The TiO2-SnS2 nanostrcuture exhibits lower reflection than the bare TiO2 at the visible light, which comes from the increased light absorption by SnS2 nanoparticles”

  Reply: Thanks to the reviewer for this helpful comment. We have added references to that statement. The specific additions are as follows:

  “The TiO2-SnS2 nanostrcuture exhibits lower reflection than the bare TiO2 at the visible light, which comes from the increased light absorption by SnS2 nanoparticles [29].”

Reference:

[29] Juan Gao, Xiaowei Sun, Lingcheng Zheng, Gang He, Yanfen Wang, Yang Li, Yin Liu, Jiale Deng, Mei Liu, 2D Z-scheme TiO2/SnS2 Heterojunctions with Enhanced Visible-light Photocatalytic Performance for Refractory Contaminants and Mechanistic Insight, New Journal of Chemistry 35 (2021).

5. Does the rough surface of the TiO2 affect the sensing performance?

  Reply: Thanks for reviewer’s helpful comment. The surface roughness of TiO2 does affect the humidity sensing performance. The specific additions are as follows:

  “And the roughness of the TiO2 surface will also affect the performance of the humidity sensor, which can be expressed as the larger the specific surface area of TiO2 the better the effect of the humidity sensor, and conversely the smaller the specific surface area of TiO2 the worse the effect of the humidity sensor [43].”

Reference:

[43] Zhao, X. Chen, X. Ding, X. Yu, X. Chen and F. Liu, Humidity Sensing Properties and Negative Differential Resistance Effects of TiO2 Nanowires, IEEE Sensors Journal, 17 (2021) 18477-18482.

6. Some grammar errors in the manuscript need to be corrected.

  Reply: We thank the reviewers for their helpful suggestions. We apologize for our errors. The manuscript has been carefully checked and the grammatical errors have been corrected.

Round 2

Reviewer 1 Report

All comments provided by me in the first review were included by the Authors in the revised version of the article, which I accepted. I think that in its current form there are no contraindications for publishing this article.